# Dimeric R25CPTH(1–34) activates the parathyroid hormone-1 receptor in vitro and stimulates bone formation in osteoporotic female mice

Minsoo Noh[1,2†], Xiangguo Che[3†], Xian Jin[3], Dong-Kyo Lee[3], Hyun-Ju Kim[3], Doo Ri Park[4], Soo Young Lee[4], Hunsang Lee[2], Thomas J Gardella[5], Je-Yong Choi[3*], Sihoon Lee[1*]

[1]Department of Internal Medicine and Laboratory of Genomics and Translational Medicine, Gachon University College of Medicine, Incheon, Republic of Korea; [2]Department of Life Sciences, Korea University, Seoul, Republic of Korea; [3]Department of Biochemistry and Cell Biology, Cell and Matrix Research Institute, School of Medicine, Kyungpook National University, Daegu, Republic of Korea; [4]Department of Life Sciences, Multitasking Macrophage Research Center, Ewha Womans University, Seoul, Republic of Korea; [5]Endocrine Unit, Massachusetts General Hospital and Harvard Medical School, Boston, United States

*For correspondence:
jechoi@knu.ac.kr (J-YC);
shleemd@gachon.ac.kr (SL)

†These authors contributed
equally to this work

## eLife Assessment

This work investigates the functional difference between the most commonly expressed form of PTH, and a mutant form of PTH, identified in a patient with chronic hypocalcemia and hyperphosphatemia which characterizes hypoparathyroidism. The authors investigate the hypothesis that this mutant PTH assumes a dimeric form in vivo and serves anabolic functions in the bone. The data are **compelling** and the translational aspects are **fundamental** in understanding PTH-1 Receptor activation.

**Abstract** Osteoporosis, characterized by reduced bone density and strength, increases fracture risk, pain, and limits mobility. Established therapies of parathyroid hormone (PTH) analogs effectively promote bone formation and reduce fractures in severe osteoporosis, but their use is limited by potential adverse effects. In the pursuit of safer osteoporosis treatments, we investigated R25CPTH, a PTH variant wherein the native arginine at position 25 is substituted by cysteine. These studies were prompted by our finding of high bone mineral density in a hypoparathyroidism patient with the R25C homozygous mutation, and we explored its effects on PTH type-1 receptor (PTH1R) signaling in cells and bone metabolism in mice. Our findings indicate that R25CPTH(1–84) forms dimers both intracellularly and extracellularly, and the synthetic dimeric peptide, R25CPTH(1–34), exhibits altered activity in PTH1R-mediated cyclic AMP (cAMP) response. Upon a single injection in mice, dimeric R25CPTH(1–34) induced acute calcemic and phosphaturic responses comparable to PTH(1–34). Furthermore, repeated daily injections increased calvarial bone thickness in intact mice and improved trabecular and cortical bone parameters in ovariectomized (OVX) mice, akin to PTH(1–34). The overall results reveal a capacity of a dimeric PTH peptide ligand to activate the PTH1R in vitro and in vivo as PTH, suggesting a potential path of therapeutic PTH analog development.

## Introduction

Osteoporosis is a prevalent global bone disorder characterized by low bone mineral density (BMD), causing weakened bones leading to fragility fractures, particularly in the spine, hip, and wrist. The development of osteoporosis is influenced by factors including gender, being more prevalent in women; hormonal changes, like decreased estrogen levels during menopause; and age, with heightened susceptibility postmenopause in women contributing to bone loss. Recent meta-analysis of previous studies indicates a global osteoporosis prevalence of 23.1% among women and 11.7% among men (*Sözen et al., 2017*; *Salari et al., 2021*). Osteoporosis stands as a noteworthy risk factor over the age of 50 years that poses challenges to the preservation of autonomous mobility and overall well-being within an aging society. There is thus a pressing need for safe and efficacious therapies for osteoporosis that mitigate fractures, alleviate associated symptoms, and preserve physical functionality.

Antiresorptive agents (e.g. bisphosphonates and denosumab) and romosozumab encompass a therapeutic approach that aims to specifically counteract the declines in bone mass by tempering the balance between bone resorption and formation (*Papapoulos et al., 2012*; *Reid et al., 2009*; *McClung et al., 2014*). It is pertinent to acknowledge, however, that the prolonged use of most of such agents is limited due to potential long-term side effects. Furthermore, antiresorptive therapies cannot stimulate new bone formation (*Cosman et al., 2016*). In contrast, bone anabolic agents, such as parathyroid hormone (PTH) and its analogs, such as teriparatide (recombinant human PTH(1–34)), increase BMD by stimulating bone formation more than bone resorption (*Martin et al., 2021*). PTH has an exceptionally short half-life in the blood of approximately 2–4 min (*Bieglmayer et al., 2002*; *Maier et al., 1998*), which helps in avoiding excessive increases in blood calcium levels that can otherwise limit the utility of PTH-related medications, while yet inducing a desired anabolic effect on bone. It is also worth noting that studies in rodents reveal that long-term administration of a PTH anabolic agent can lead to bone overgrowth, osteosarcoma, as well as hypercalcemia (*Vahle et al., 2002*; *Appelman-Dijkstra and Papapoulos, 2016*). Consequently, a goal of ongoing research is to uncover the underlying molecular mechanisms driving the anabolic and catabolic effects of PTH to thereby secure more effective therapeutic alternatives for osteoporosis (*Khosla and Hofbauer, 2017*).

PTH is produced and secreted by the parathyroid glands as a straight-chain monomeric polypeptide of 84 amino acids (*Moallem et al., 1998*; *Silver et al., 1985*; *Chen and Goodman, 2004*). It plays a vital role in maintaining calcium and phosphate equilibrium by acting on the PTH1R, a class B G protein-coupled receptor (*Habener et al., 1984*; *Veldurthy et al., 2016*; *Kiela and Ghishan, 2009*) that is expressed primarily in cells of bone and kidney (*Jüppner et al., 1991*). The orchestrated downstream effects of PTH in target cells act to ensure optimal bone health and the maintenance of the ambient blood calcium and phosphate concentrations required for proper nerve conduction, muscle activity, and systemic cellular communication, whereas disturbances in this system can lead to multiple disorders.

Central to the PTH signaling cascade is the activation of intracellular G proteins (*Gardella and Jüppner, 2000*), most prominently Gsα, which in turn activates transmembrane adenylyl cyclases, leading to the synthesis of the second messenger cyclic AMP (cAMP), and the activation of cAMP-dependent protein kinase A (PKA) (*Weinstein et al., 2001*; *Feinstein et al., 2011*). PTH can also activate other second messenger cascades, including the Gαq/phospholipase C/inositol 1,4,5-trisphosphate, diacylglycerol, and protein kinase C signaling pathways (*Iida-Klein et al., 1997*; *Dunlay and Hruska, 1990*), highlighting the diverse biology of PTH and the PTH1R (*Abou-Samra et al., 1992*). Adding to this paradigm, the PTH1R also mediates the actions of parathyroid hormone-related protein (PTHrP), a development protein that acts in the formation of bones and other tissues. PTHrP shares considerable sequence homology with PTH in the first 34 amino acids, which encompass the receptor-binding portions of the two respective ligands. The PTH1R thus has an intrinsic capacity for dual ligand recognition, which opens possibilities for exploring new modes of therapeutic development for diseases such as osteoporosis and hypoparathyroidism (*Jüppner et al., 1991*; *Gardella and Jüppner, 2000*; *Bone et al., 2004*; *Rubin et al., 2016*; *Winer et al., 1996*; *Winer et al., 2018*). Pharmacologically, the PTH1R can adopt at least two distinct ligand-binding conformations, RG and $R^0$, the selectivity for which can lead to altered modes of signaling in vitro and in vivo for peptides such as PTH, PTHrP, and various hybrid analogs (*Okazaki et al., 2008*; *Pioszak et al., 2009*; *Cheloha et al., 2015*; *Hoare et al., 2001*; *Dean et al., 2008*).

The current study extends our prior investigation in which we identified in a patient with chronic hypocalcemia and hyperphosphatemia with a mutation that changes the arginine at position 25 in the mature PTH(1–84) polypeptide to cysteine (R25CPTH) (*Lee et al., 2015*; *Bae et al., 2016*). Antibody assays revealed the R25CPTH mutant protein to be present in the patient's blood at markedly elevated levels. We now have found that this patient expressing the R25CPTH variant has higher-than-normal BMD for age and sex. We further characterize the R25CPTH protein and find that it can manifest in two distinct molecular forms: as a monomer and as a dimer, and we demonstrate that a dimeric R25CPTH(1–34) synthetic peptide retains agonistic properties on the PTH1R that are driven by a moderate selectivity for the RG versus R0 receptor conformation. Finally, we demonstrate in mice that R25CPTH(1–34) can induce skeletal responses that are similar to those induced by PTH(1–34), but without triggering an excessive hypercalcemic response. Considering the proven bone anabolic capacity of several established PTH agonist ligands, and the need for safe, long-term treatments for skeletal disorders, our studies on dimeric R25CPTH(1–34) suggest alternative strategies to consider in such drug development programs.

## Results

### Dimerization of R25CPTH(1–84)

In our previous studies, we showed that synthetic monomeric peptide, R25CPTH(1–34), as compared to PTH(1–34), exhibits a moderately diminished PTH1R-binding affinity and decreased cAMP signaling potency in vitro, and that with long-term infusion in mice, R25CPTH(1–34) leads to only minimal calcemic and phosphaturic effects, which corroborates the hypocalcemia seen in the original patient, despite the significantly elevated levels of R25CPTH in the plasma (*Lee et al., 2015*). In subsequent analysis of this patient, we found a particularly high BMD for age and sex (*Figure 1—figure supplement 1*), which prompted us to further characterize the functional properties of R25CPTH, as described herein. We produced recombinant PTH(1–84) with or without the R25C mutation by expression of the corresponding cDNA in HEK293T cells (*Figure 1*). We considered the possibility that the introduction of a sole new cysteine within the polypeptide chain of R25CPTH(1–84) might induce homologous bimolecular dimerization through a disulfide bond involving the thiol functional group in each monomer (*Banerjee and Lazar, 2001*; *Trivedi et al., 2009*). To specifically investigate this, we designed the cDNA constructs to express either pre-pro-PTH(1–115)-3xFLAG or R56Cpre-pro-PTH(1–115)-3xFLAG, such that after intracellular processing and cleavage of the pre-pro regions, the mature PTH(1–84)-3xFLAG or R25CPTH(1–84)-3xFLAG peptides would be generated upon transfection in HEK293T cells (*Figure 1C*; *Kemper et al., 1974*; *Vasicek et al., 1983*; *Wiren et al., 1989*). We performed western blot analysis of total cell lysates and conditioned culture media collected from the transfected cells to specifically assess the possible presence of a disulfide-bonded dimeric form. Each sample was thus prepared in either reduced or non-reduced form and proteins were detected using an anti-flag antibody. The results demonstrated the presence of both a low molecular weight monomeric, and in the non-reduced samples, a higher molecular weight dimeric form of the R25CPTH(1–84)-3XFLAG protein in both the cell lysate and extracellular conditioned medium fractions, and the dimer appeared to be at an elevated proportion in the medium relative to the lysate (*Figure 1D*). To control for potential artifact effects attributed to the 3xFLAG tag, we utilized plasmid constructs pcDNA3.0-(pre-pro-PTH)-IRES and pcDNA3.0-(R56Cpre-pro-PTH)-IRES encoding non-tagged PTH variants and an anti-PTH(39–84) antibody for western blot analysis, which again confirmed the presence of the dimer in total HEK293T cell lysates (*Figure 1—figure supplement 2*).

We observed in the above studies that the expression level of R25CPTH(1–84) was higher than that of wild-type PTH(1–84) in both cell lysate and medium, which we considered might be due to an intrinsic enhancement in protein stability and resistance to protein degradation in the dimeric molecule. To address this, we treated the cells with the proteasome inhibitor MG132, which acts by forming a hemiacetal with the hydroxyl groups of active site threonine residues, and compared the expression levels of wild-type PTH(1–84) and R25CPTH(1–84) in the treated versus untreated cells. The results indicated that while the expression level of wild-type PTH was increased by MG132 treatment, it did not reach the level of R25CPTH, suggesting that the difference in expression is not related to a difference in sensitivity to proteasome-mediated degradation (*Figure 1—figure supplement 3*).

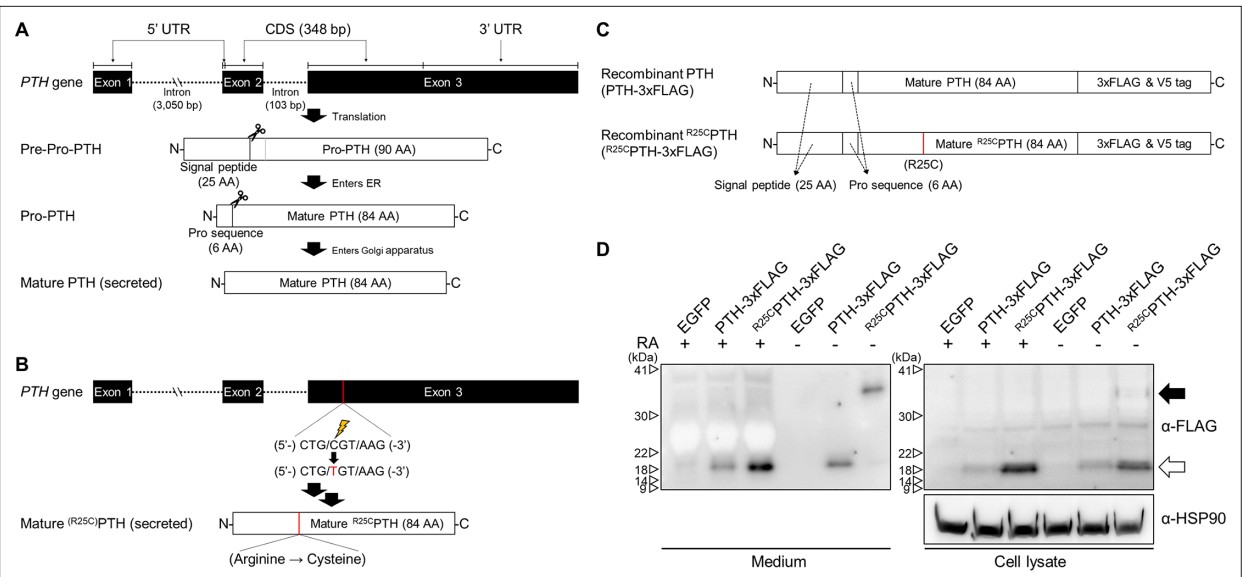

**Figure 1.** Formation of R25C mutant PTH(1–84) dimer. (**A**) Schematic representation of PTH gene structure and expression. (**B**) Schematic representation of [R56C]pre-pro-PTH(1–115) (in mature form, [R25C]PTH(1–84)) gene structure and expression. (**C**) Schematic representation of recombinant PTH proteins. (**D**) In vitro dimerization of [R25C]PTH. Recombinant protein constructs were transfected into HEK293T cells, and expression of PTH-3xFLAG and [R25C]PTH3xFLAG in culture medium or cell lysate was demonstrated by western blot. The result confirms the presence of dimeric [R25C]PTH (*bp: base pairs; *AA: amino acids; *RA: reducing agent).

The online version of this article includes the following source data and figure supplement(s) for figure 1:

**Source data 1.** The original files of the raw membranes correspond to *Figure 1D*.

**Source data 2.** The uncropped membranes correspond to *Figure 1D* indicating the relevant bands and treatments.

**Figure supplement 1.** Bone mineral density (BMD) analysis in a patient affected by homogeneous [R25C]PTH mutation.

**Figure supplement 2.** Identification of dimeric [R25C]PTH (1–84) peptide.

**Figure supplement 2—source data 1.** The original files of the raw membranes correspond to *Figure 1—figure supplement 2*.

**Figure supplement 2—source data 2.** The uncropped membranes correspond to *Figure 1—figure supplement 2* indicating the relevant bands and treatments.

**Figure supplement 3.** Influence of proteasome inhibitor MG132 on PTH and [R25C]PTH stability.

**Figure supplement 3—source data 1.** The original files of the raw membranes correspond to *Figure 1—figure supplement 3*.

**Figure supplement 3—source data 2.** The uncropped membranes correspond to *Figure 1—figure supplement 3* indicating the relevant bands and treatments.

Overall, we have confirmed that [R25C]PTH(1–84) can form a dimeric structure, and the [R25C]PTH(1–84) secreted outside the cells predominantly exists in dimeric form. Thus, utilizing dimer [R25C]PTH(1–84) in the analysis would be more relevant to understanding the actual function of [R25C]PTH. Consequently, we aim to conduct further validation using dimeric [R25C]PTH in our subsequent investigations.

## Functional characterization of dimeric [R25C]PTH(1–34) in vitro

To explore the functional properties of dimeric [R25C]PTH, we conducted experiments using synthetic peptides of PTH(1–34), [R25C]PTH(1–34) (monomeric), and disulfide-bonded dimeric [R25C]PTH(1–34). First, we examined the receptor-binding affinity of these ligands by performing competition experiments using membranes prepared from HEK293-derived GP-2.3 cells that stably express the human PTH1R and assay formats designed to assess binding to either the G protein-uncoupled $R^0$ or G protein-coupled RG receptor conformation. The results revealed that monomeric [R25C]PTH(1–34) bound to both the $R^0$ and RG conformations with comparable, albeit slightly weaker affinity as compared to PTH(1–34), while dimeric [R25C]PTH(1–34) bound to each conformation with weaker affinity than did the monomeric form while showing an apparent selectivity for higher binding to the RG versus $R^0$ conformation of PTH1R (*Figure 2A*).

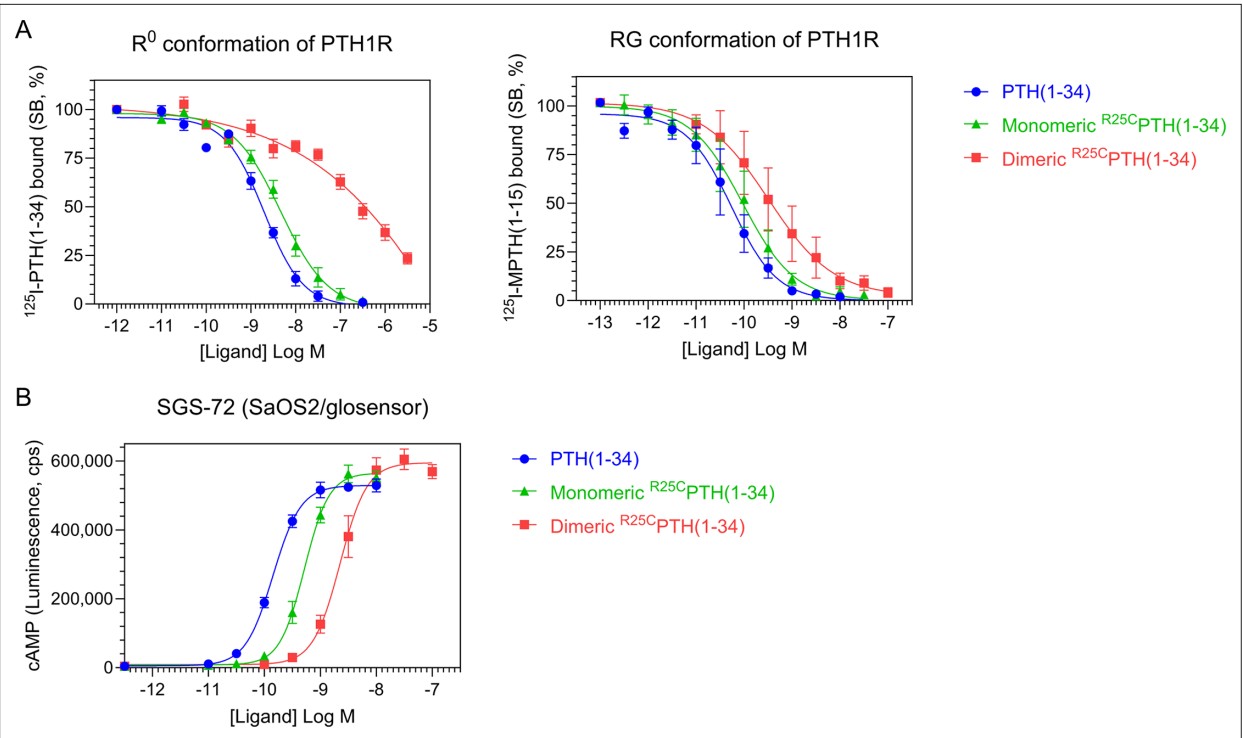

**Figure 2.** Effect of PTH, monomeric [R25C]PTH, and dimeric [R25C]PTH to the PTH1R in vitro. (**A**) The binding of PTH(1–34), monomeric [R25C]PTH(1–34), and dimeric [R25C]PTH(1–34) to the PTH1R in $R^0$ conformation of RG conformation was assessed by competition methods using [125I]-PTH(1–34) and [125I]-MPTH(1–15) as radioligand (n=4). (**B**) Ligand potency for cyclic AMP (cAMP) signaling was assessed in SGS-72 cells, which were derived from SaOS-2 cells modified to express Glosensor cAMP reporter (n=4). The cells were preloaded with luciferin and treated with varying concentrations of PTH(1–34), monomeric [R25C]PTH(1-34), and dimeric [R25C]PTH(1–34). Error bars represent mean ± standard error.

To investigate the signaling properties of the ligands, we measured the changes in the increase in intracellular levels of cAMP induced by each ligand in an osteoblastic SaOS-2-derived cell line (SGS-72 cells) that stably expresses the Glosensor cAMP reporter. These assays revealed that each ligand dose-dependently increased the cAMP levels in the cells, detected as an increase in luminescence in an Envision plate reader, and while the potencies were moderately and more substantially reduced for the monomeric and dimeric forms of the ligand, respectively, as compared to PTH(1–34), the maximum response attained by each ligand was comparable (*Figure 2B*). Dimeric [R25C]PTH(1–34) thus retains signaling functionality at the PTH1R that is characterized by a potency approximately commensurate with its affinity for binding to the RG PTH1R conformation.

### Effect of single injection of dimeric [R25C]PTH(1–34) on calcium and phosphate regulation in mice

To assess whether dimeric [R25C]PTH can function in vivo, we injected the ligand, and in parallel either vehicle or PTH(1–34) (each peptide at a dose of 50 nmol/kg) into CD1 female mice and measured levels of ionized calcium ($Ca^{2+}$) in the blood (n=6 mice/group), inorganic phosphate (Pi) in plasma (n=12 mice/group), and the excretion of Pi into urine (n=6 mice/group). Blood $Ca^{2+}$ levels were measured at serial time points of pre-injection, 1, 2, 4, and 6 hr post-injection. Both PTH(1–34) and dimeric [R25C]PTH(1–34) induced increases in blood $Ca^{2+}$ levels that were significant, relative to the levels in vehicle-injected mice, at 1 and 2 hr post-injection, and the levels then returned to the baseline levels of vehicle control mice by 4 hr (*Figure 3A*). Plasma Pi levels were measured in samples obtained pre-injection, at 6 min, and 1, 2, and 6 hr post-injection. PTH(1–34) induced a significant decrease in plasma Pi at 1 hr post-injection, and the levels subsequently returned to baseline by 2 hr. Injection of dimeric [R25C]PTH(1–34) resulted in a slight decrease in plasma Pi at 2 hr post-injection, but the effect was not significant (*Figure 3B*). Consistent with this trend, Pi levels in the urine of mice injected with dimeric [R25C]PTH(1–34) were increased significantly at 2 hr post-injection and then gradually returned

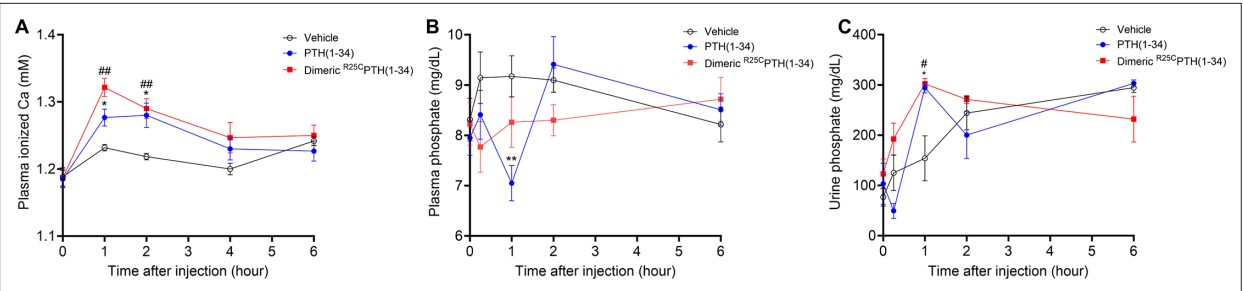

**Figure 3.** Calcemic and phosphatemic responses by PTH injection in CD1 female mice. (**A**) Plasma calcemic response after injection (n=6). Both PTH(1–34) and dimeric $^{R25C}$PTH(1–34) significantly elevate ionized calcium levels in plasma at 1–2 hr post-injection. After 2 hr post-injection, plasma ionized calcium level gradually restored to baseline levels similar to those of the vehicle group. (**B**) Plasma phosphatemic response after injection (n=12). Following PTH(1–34) injection, plasma phosphate levels significantly decrease at 1 hr post-injection, subsequently returning to baseline akin to those of the vehicle group. Conversely, dimeric $^{R25C}$PTH(1–34) injection shows no significant alteration in phosphatemic response but demonstrates a tendency toward a slight decrease in phosphate levels, gradually restoring to baseline levels akin to those of the vehicle group. (**C**) Urine phosphatemic response after injection (n=6). The urine phosphate levels markedly increased at 1 hr post-injection for both PTH(1–34) and dimeric $^{R25C}$PTH(1–34), followed by a return to baseline levels akin to those of the vehicle group. This analysis was conducted using 9-week-old female CD1 mice. The mice were administered PTH(1–34) and dimeric $^{R25C}$PTH(1–34) at a concentration of 50 nmol/kg for each compound. Error bars represent mean ± standard error. p-Values were determined using the two-way ANOVA to compare the mean of each test cell with the mean of the control cell at the same time point. * denotes p-value<0.05 for PTH(1–34) compared to vehicle, ** denotes p-value<0.01 for PTH(1–34) compared to vehicle, # denotes p-value<0.05 for dimeric $^{R25C}$PTH(1–34) compared to vehicle, ## denotes p-value<0.01 for dimeric $^{R25C}$PTH(1–34) compared to vehicle.

to baseline levels (*Figure 3C*). These results thus indicate that dimeric $^{R25C}$PTH can elicit calcemic and phosphaturic responses in vivo that are fully with those expected for an injected PTH1R agonist ligand.

## Effect of dimeric $^{R25C}$PTH(1–34) on bone calvariae in mice

To initially assess the effects that short-term treatment of dimeric $^{R25C}$PTH(1–34) can have on bone, we injected 8-week-old male C57BL/6 mice once a day for 6 days (days 1–6), with either dimeric $^{R25C}$PTH(1–34), PTH(1–34) or vehicle, and after 10 days without treatment (day 16) followed by euthanasia, we isolated the calvariae for histological analysis of new bone formation. Specifically, we examined sections stained with hematoxylin and eosin (H&E) to assess the width of newly formed bone areas along the edge of each sample. These regions exhibited a more vivid coloration compared to the surrounding existing bone tissue, demarcated by a dotted line for clarity (*Figure 4A*). Measurements were taken below and above the dissection area where new bone had formed, and these measurements were then utilized to calculate the mean values for further analysis. These analyses revealed that both PTH(1–34) and dimeric $^{R25C}$PTH(1–34) significantly increased the width of the new bone area by approximately fourfold, as compared to the vehicle group (*Figure 4B*). These findings thus support a capacity of dimeric $^{R25C}$PTH(1–34) to induce new bone formation in vivo, similar to PTH, despite molecular and structural changes.

## Effect of dimeric $^{R25C}$PTH on bone mass in osteoporotic mice

To more directly assess the impact of dimeric $^{R25C}$PTH(1–34) on bone mass, we administered it to OVX mice, which serve as a well-established model for postmenopausal osteoporosis. The OVX mice were injected five times per week for 4 weeks with either the dimeric ligand (OVX+dimeric $^{R25C}$PTH(1–34)), PTH(1–34) (OVX+PTH(1–34)), or vehicle (OVX+vehicle, OVX-controls), and sham-operated (Sham) mice were used as further controls. Mice were euthanized at the end of the injection period and tissue samples were isolated for analysis. Quantitative micro-computed tomography (μ-CT) analysis of the femurs obtained from each group revealed that, as compared to OVX+vehicle controls, treatment with PTH(1–34) increased femoral trabecular bone volume fraction (Tb.BV/TV) by 121%, cortical bone volume fraction (Ct.BV/TV) by 128%, cortical thickness (Ct.Th) by 115%, cortical area (Ct.Ar) by 110%, and cortical area fraction (Ct.Ar/Tt.Ar) by 118%, while decreased total tissue area (Tt.Ar) by 93% (*Figure 5A and B*). Treatment with dimeric $^{R25C}$PTH(1–34) had similar effects on the femoral cortical bone parameters, as it increased Ct.BMD by 104%, Ct.BV/TV by 125%, Ct.Th by 107%, and Ct.Ar/Tt.Ar by 116%, while decreased Tt.Ar 86% (*Figure 5*). Considering the reduction of Tt.Ar and no change of Ct.Ar compared to the OVX+vehicle controls, the increase of Ct.Ar/Tt.Ar indicates a decrease in bone

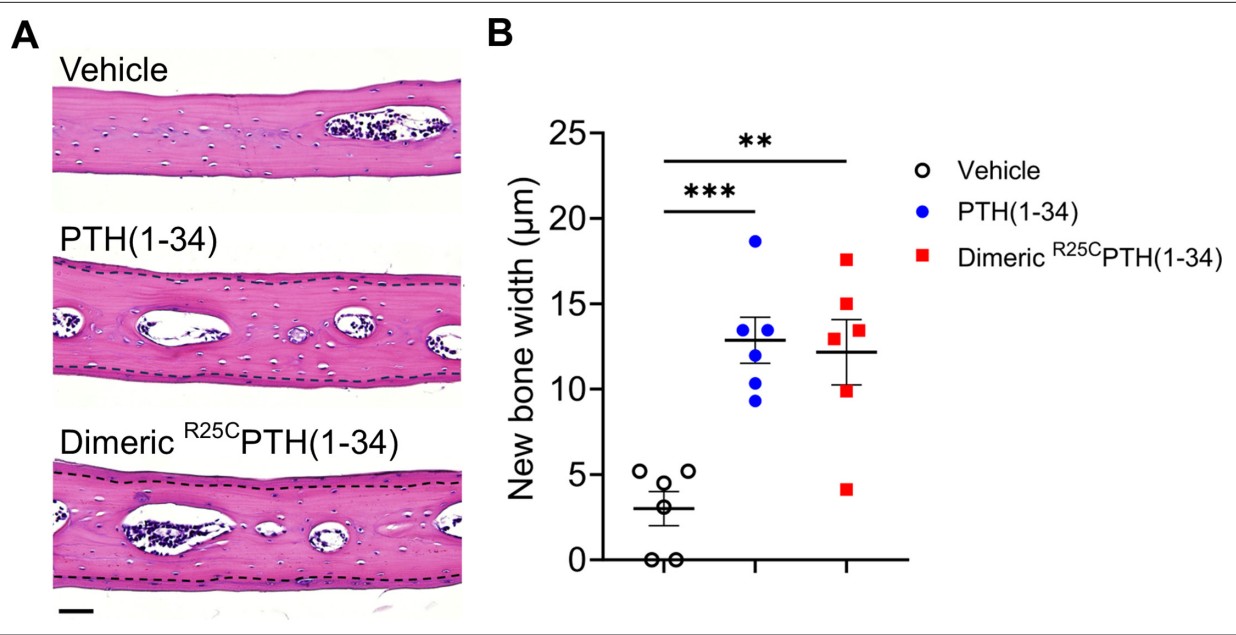

**Figure 4.** Effect of dimeric [R25C]PTH(1–34) in calvarial injection model. (**A**) Dissections of the calvarial bones. Calvarial injections were performed on 8-week-old male C57BL/6 mice (N=6 per group) that received daily administrations of vehicle, PTH(1–34), or dimeric [R25C]PTH(1–34) for 6 days. Following a 10-day treatment period, histological sections of calvariae, stained with hematoxylin (blue-purple, indicating cell nuclei) and eosin (pink, representing bone matrix), were obtained. The area of new bone formation, with more intense staining compared to the existing bone tissue, is denoted by the dotted line. Scale bar = 50 μm. (**B**) Quantification of new bone width in calvarial injection model. The result showed a significant increase in new bone width following injections of both PTH(1–34) and dimeric [R25C]PTH(1–34) compared to the vehicle group. p-Values were obtained using the one-way ANOVA to compare the mean of each column with the mean of a control column. ** indicates p-value<0.01 against vehicle, *** indicates p-value<0.001 against vehicle.

marrow space. The increase in cortical bone BMD was significant with dimeric [R25C]PTH(1–34) but not with PTH(1–34), whereas an increase in femoral trabecular bone was only observed with PTH(1–34).

The effects of the treatments on bone strength were assessed by conducting a three-point bending test on femurs isolated from the mice. The maximum load parameter was significantly decreased in the femurs from OVX-control versus Sham mice and was significantly increased, relative to the OVX-controls, by treatment with the PTH(1–34), but not dimeric [R25C]PTH(1–34) (***Figure 5C***). The slope parameter was significantly decreased in the femurs from OVX-control versus Sham mice and tended to increase versus OVX-controls by treatment with either PTH(1–34) or [R25C]PTH(1–34), but the changes were not significant.

We further analyzed the levels of bone metabolism markers in the serum obtained from the mice at the study endpoint (***Figure 5D***). The levels of serum calcium were within the normal range in all treatment groups, while serum phosphate levels were modestly increased in the OVX mice treated with PTH(1–34) or dimeric [R25C]PTH(1–34) as compared to with vehicle, but the effect was significant only with PTH(1–34). The serum levels of CTX-1, a bone resorption marker (***Rosen et al., 2000***), were elevated in each of the OVX groups versus the Sham group, and tended to be lower in the OVX-PTH(1–34) treatment group, but the change relative to OVX-vehicle was not significant (***Figure 5D***). Interestingly, serum P1NP and alkaline phosphatase (ALP) levels were significantly increased in dimeric [R25C]PTH(1–34)-treated group, compared to the OVX-vehicle group (***Pagani et al., 2005***; ***Wheater et al., 2013***). Histological staining of proximal tibial sections for tartrate-resistant acid phosphatase (TRAP) activity, a marker of osteoclast-mediated bone resorption, revealed an apparent increase in this activity in bones of the OVX mice, as compared to those in Sham-control mice, reflecting a heightened rate of bone turnover, as also suggested by the increased levels of serum CTX-1 in the OVX mice, and the TRAP staining appeared to be reduced in the tibiae of the OVX mice treated with PTH(1–34) (***Figure 5D and E***). Further histomorphometric analysis confirmed a significant increase in the osteoclast surface area relative to bone surface area (Oc.S/BS) in the proximal tibiae of the OVX-vehicle

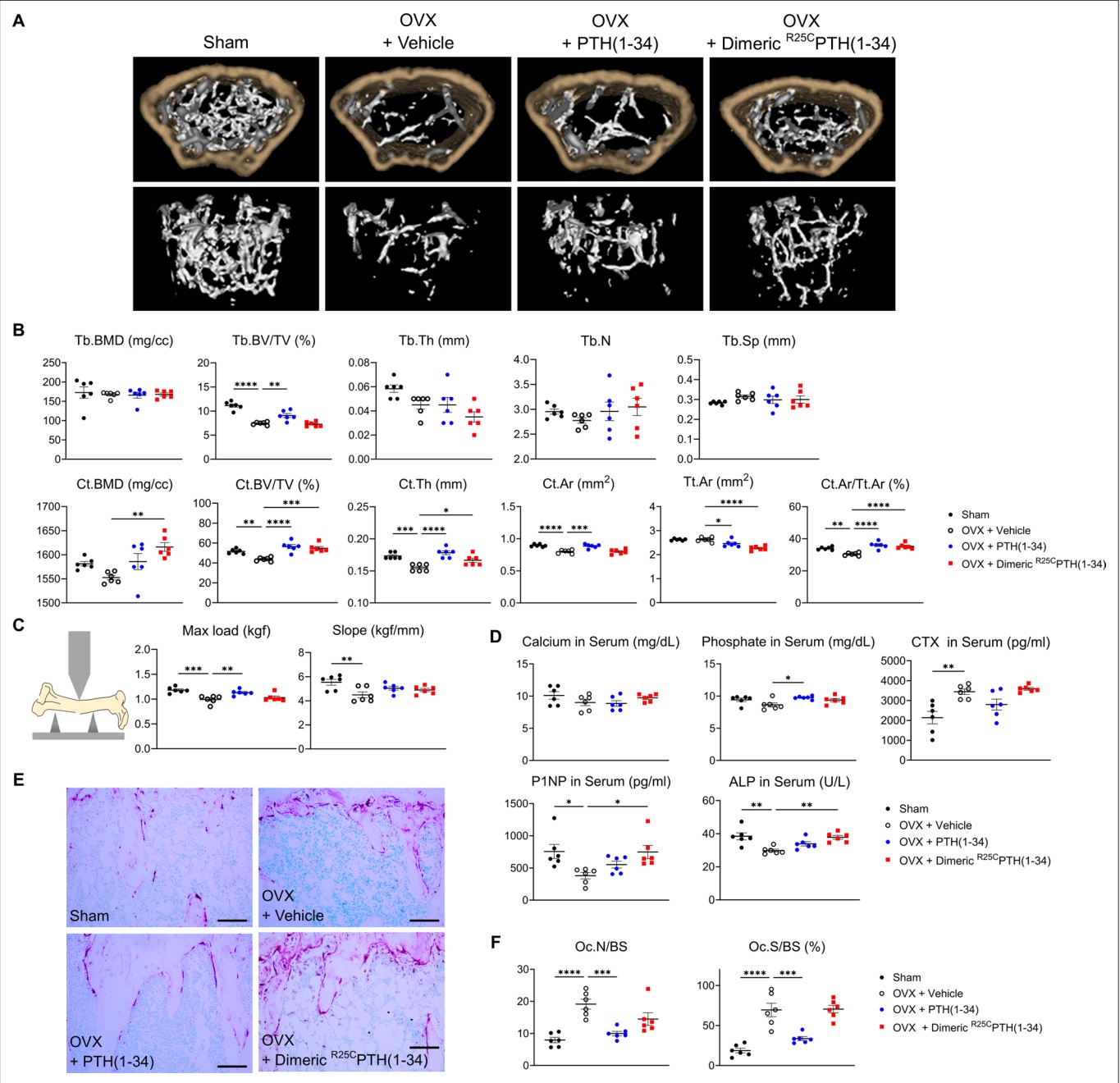

**Figure 5.** Impact of [R25C]PTH(1–34) on bone turnover. The effects of Sham, OVX-control (OVX+vehicle), OVX treated with PTH(1–34) (OVX+PTH(1–34)), and OVX treated with dimeric [R25C]PTH(1–34) (OVX+dimeric [R25C]PTH(1–34)) on bone turnover in mice. (**A**) Femurs obtained from mice in each group were subjected to micro-computed tomography (μ-CT) analyses for the assessment of bone mass. (**B**) Several parameters of (**A**) were quantified using μ-CT measurements, including trabecular bone mineral density (Tb.BMD), trabecular bone volume to tissue volume (Tb.BV/TV), trabecular bone thickness (Tb.Th), trabecular number (Tb.N), trabecular separation (Tb.Sp), cortical bone mineral density (Ct.BMD), cortical bone volume to tissue volume (Ct. BV/TV), cortical thickness (Ct.Th), cortical area (Ct.Ar), total tissue area (Tt.Ar), and cortical area to total tissue area (Ct.Ar/Tt.Ar). (**C**) A 3D-point bending test was conducted with femurs obtained from mice in each group. The left panel describes a schematic model of a 3D-point bending test. The middle and right panels each indicate the maximum bending load (kgf) and slope (kgf/mm). (**D**) Serum levels of calcium, phosphorus, CTX, P1NP, and alkaline phosphatase (ALP) were measured for each group using an enzyme-linked immunosorbent assay (ELISA). (**E**) Tartrate-resistant acid phosphatase (TRAP) staining of histological sections of proximal tibias was carried out to visualize osteoclast activity. Scale bars = 100 μm. (**F**) Quantification of osteoclast number per bone surface (Oc.N/BS) and osteoclast surface per bone surface (Oc.S/BS) was performed. Each group consisted of six samples (n=6). The error bars indicate mean ± standard error. p-Values were obtained using the one-way ANOVA to compare the mean of each column with the mean of a control column. * indicates p-value<0.05, ** indicates p-value<0.01, *** indicates p-value<0.001, **** indicates p-value<0.0001.

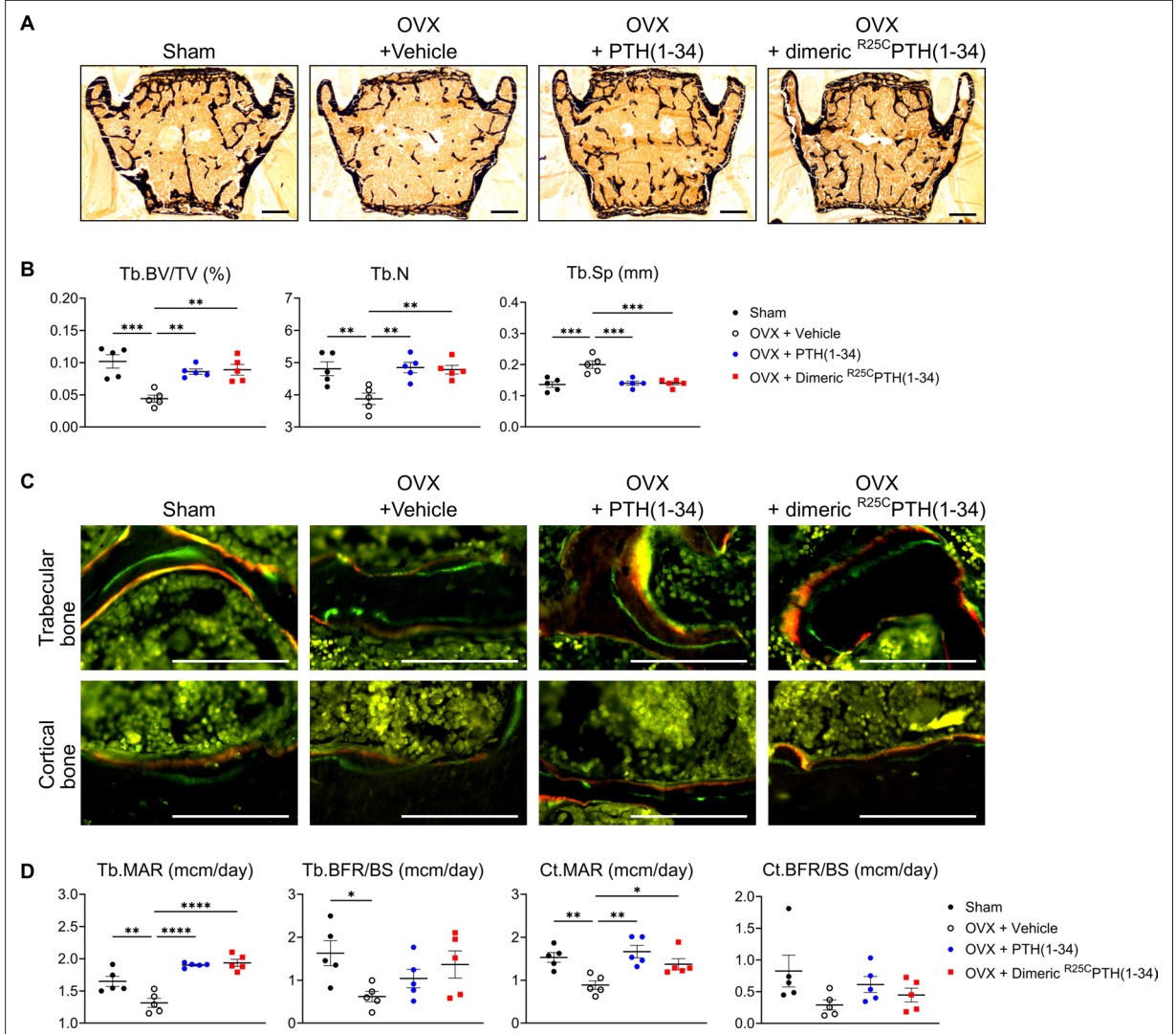

**Figure 6.** Impact of [R25C]PTH(1–34) on osteoblast function in vertebrae. The effects of Sham, OVX-control (OVX+vehicle), OVX treated with PTH(1–34) (OVX+PTH(1–34)), and OVX treated with dimeric [R25C]PTH(1–34) (OVX+dimeric [R25C]PTH(1–34)) on osteoblast function in mice. (**A**) Mineralization of vertebrae obtained from each group was assessed through von Kossa staining. Scale bars = 100 μm. (**B**) Quantification of trabecular bone parameters including trabecular bone volume to tissue volume (Tb.BV/TV), trabecular number (Tb.N), and trabecular separation (Tb.Sp) was performed using the Bioquant Osteo 2019 v19.9.60 program. (**C**) Fluorescent microscopic observations of trabecular and cortical bone sections from each group demonstrate the apposition of xylenol (red) and calcein (green) labels. Scale bars = 100 μm. (**D**) Quantification of trabecular bone parameters such as trabecular bone mineral apposition rate (Tb.MAR), trabecular bone formation rate to bone surface (Tb.BFR/BS), cortical bone MAR (Ct.MAR), and trabecular MAR (Tb.MAR) was carried out using the Bioquant Osteo 2019 v19.9.60 program. Each group consisted of five samples (n=5). The error bars indicate mean ± standard error. p-Values were obtained using the one-way ANOVA to compare the mean of each column with the mean of a control column. * indicates p-value<0.05, ** indicates p-value<0.01, *** indicates p-value<0.001, **** indicates p-value<0.0001.

mice, relative to that in the Sham-control mice, and this parameter was significantly decreased by treatment with PTH(1–34), but not with dimeric [R25C]PTH(1–34) (*Figure 5F*).

We then analyzed the bone microstructure in the lumbar vertebrae through von Kossa staining of histological sections and histomorphometric quantification (*Figure 6A*). The trabecular bone volume fraction (Tb.BV/TV, %) and trabecular number (Tb.N) were significantly reduced in the OVX-vehicle group, as compared to the Sham group, and treatment with either PTH(1–34) or dimeric [R25C]PTH(1–34) resulted in a significant increase in each of these parameters, as well as a concomitant reduction in trabecular separation (Tb.Sp), as compared to the respective parameters in the OVX-vehicle group (*Figure 6B*).

Dynamic bone histomorphometry was also performed on the vertebrae to evaluate rates of bone formation (*Figure 6C*). The trabecular mineral apposition rate (MAR) and cortical MAR were significantly increased in both the OVX-PTH(1–34) and OVX-dimeric [R25C]PTH(1–34) groups, as compared to in the OVX-vehicle group, and although there was a tendency for an increase in the bone formation rate (BFR/BS) in both the trabecular and cortical bone with either PTH(1–34) or dimeric [R25C]PTH(1–34) treatment, the differences were not statistically significant, as compared to the OVX-control (*Figure 6D*).

In summary, injection of dimeric [R25C]PTH into osteoporotic OVX mice resulted in significant increases in cortical bone in the femurs and trabecular bone in the vertebrae, as well as significant increases in osteoblast function and serum markers of bone formation, ALP and P1NP, without inducing excessive bone resorption or hypercalcemia.

## Discussion

In this study, we show the introduction of a cysteine mutation at the 25th amino acid position of mature parathyroid hormone ([R25C]PTH) facilitates the formation of homodimers comprised of the resulting dimeric [R25C]PTH peptide in vitro. This dimerization was surprisingly compatible with receptor-binding affinity and led to relatively minor deviations in functional behavior as assessed in our cell-based assays and compared to the standard monomeric control PTH peptide. The homozygous [R25C]PTH mutation was identified in patients who presented with hypocalcemia and hyperphosphatemia, despite elevated PTH levels (*Lee et al., 2015*; *Andersen et al., 2022*), and the mutation was found to impact the bioactive region of PTH (*Lee et al., 2015*; *Arnold et al., 1990*; *Parkinson and Thakker, 1992*; *Sunthornthepvarakul et al., 1999*; *Ertl et al., 2012*). Our initial research on this [R25C]PTH mutant focused on the monomeric state of the peptide, and these studies revealed relatively moderate decreases in PTH1R-binding affinity and cAMP-stimulating potency in vitro and moderately impaired calcemic effects upon infusion in mice. Additional clinical observations, however, revealed the patients to have a higher BMD than anticipated for age-matched averages. This elevated bone mass, coupled with the elevated serum PTH levels, prompted our further investigations into the properties of the mutant PTH, as described herein.

Our investigations brought to light the capacity of the cysteine-25 mutation to induce dimer formation in the otherwise monomeric PTH polypeptide as produced in transfected cells. This result was established by comparing western blots of transfected cell lysates analyzed under reducing versus non-reducing conditions of gel electrophoresis. Subsequently, we employed synthetic peptides to further explore the functional properties of dimeric [R25C]PTH(1–34). The results of our current studies show some divergence from our previous findings obtained using the monomeric counterpart, [R25C]PTH(1–34). Compared to the monomer, dimeric [R25C]PTH(1–34) exhibited a more preferential binding affinity for the RG versus $R^0$ PTH1R conformation, despite a diminished affinity for either conformation. We also observed that the potency of cAMP production in cells was lower for dimeric [R25C]PTH as compared to the monomeric [R25C]PTH, in accordance with a lower PTH1R-binding affinity. Previous reports indicated that a mutation at the 25th position of PTH results in the loss of calcium ion allosteric effects on monomeric [R25C]PTH, leading to faster ligand dissociation, rapid receptor recycling, and a shorter cAMP time course (*White, 2019*). Correspondingly, the weaker receptor affinity and reduced cAMP production observed in dimeric [R25C]PTH suggest a possibility that the formation of a disulfide bond at the 25th position significantly alters the function of PTH as a PTH1R ligand. While these structural effects are not yet fully understood and need to be investigated further.

We further pursued in vivo applications in mice. We assessed the calcemic and phosphatemic responses to a single injection of synthetic peptides of either PTH(1–34) or dimeric [R25C]PTH(1–34) in intact mice. We found the dimer could induce increases in plasma calcium levels that were at least as robust and as sustained as those induced by PTH(1–34) and were similarly phosphaturic (*Figure 3*).

Activation of the canonical Gαs-cAMP-PKA signaling pathway is generally thought to underlie most of the biological responses induced by PTH1R activation, and our studies in SGS-72 cells confirm that dimeric PTH can activate this pathway, albeit not as efficiently as a monomeric PTH peptide. Arg25 resides in the 20–34 (C-terminal region) of PTH(1–34), which plays a significant role in the binding of the ligand to the extracellular domain (ECD) of the PTH1R. In concert with the binding of the 20–24 region of PTH to the ECD, the N-terminal portion of PTH engages the transmembrane domain of the PTH1R, inducing the conformational changes involved in G protein coupling and cAMP production (*Dean et al., 2008*; *Clark et al., 2020*; *Kumar et al., 2016*; *Zhao et al., 2019*; *Zhai et al., 2022*).

The precise binding mode used by dimeric $^{R25C}$PTH to the PTH1R is unknown, but it may be anticipated that it differs to some extent from that used by the monomeric peptide, due, for example, to the changes in bulk molecular size and display of accessible functional groups. Consequently, the receptor conformational changes and the modes of coupling to downstream effectors may differ for the monomeric versus dimeric ligands, which could potentially lead to altered signaling and biological responses in vivo. Whether such changes account for the increased bone density observed in the patient with the homozygous $^{R25C}$PTH mutation remains unknown, but cannot be presently ruled out.

The results of practical assessments of dimeric $^{R25C}$PTH(1–34) for effects on calvarial bone after short-term (6 days) injection into normal mice, and for effects on bone mass after long-term (4 weeks) daily injection in osteoporotic OVX mice demonstrate the dimer can mediate significant influences on bone metabolism alike to wild-type PTH. In OVX mice, PTH and dimeric $^{R25C}$PTH significantly increased bone formation markers P1NP and ALP. However, wild-type PTH, but not dimeric $^{R25C}$PTH, decreased bone resorption markers such as TRAP staining and CTX-1 levels, suggesting different effects of wild-type monomeric PTH and dimeric $^{R25C}$PTH. In this study, the increase of bone resorption markers of dimeric $^{R25C}$PTH compared to PTH may be one reason for low bone strength. In addition, the data showed an increase in cortical bone area (Ct.Ar) in the PTH treatment group but not in the PTH dimer treatment group. However, both dimeric and monomeric PTH treatments reduced total tissue area (Tt.Ar), suggesting potential effects on bone growth in the width of mice or humans. While further investigation is necessary, the current experimental data, along with patient symptoms showing no bone resorption despite lifelong exposure to high levels of $^{R25C}$PTH, imply that the dimeric form of $^{R25C}$PTH can provide a new insight into PTH and PTH analogs with distinct functionalities compared to the wild-type PTH.

This study has several limitations. First, it is urgently necessary to determine whether dimeric $^{R25C}$PTH is present in human patient serum. Second, TRAP staining showed an inhibitory effect of PTH treatment on the primary spongiosa area. However, the secondary spongiosa, which more accurately reflects bone remodeling (*Jilka, 2007*), was not examined due to the barely detectable bone in this area in OVX-induced osteoporosis mouse models. Third, it is unclear whether similar bone phenotypes exist in patients versus in mice treated with dimeric $^{R25C}$PTH, particularly regarding low bone strength. Although the dimeric $^{R25C}$PTH-treated group showed higher cortical BMD compared to WT-Sham or PTH groups, there was no difference in bone strength compared to the osteoporotic mouse model. Fourth, our study demonstrated that PTH or $^{R25C}$PTH treatment decreased circumferential length, which could affect bone growth in width. However, whether this phenotype is also observed in patients treated with PTH or $^{R25C}$PTH remains uncertain. Finally, we did not analyze the $^{R25C}$PTH mutant mouse model, which would allow us to compare phenotypes that most closely resemble those of human patients.

Interestingly, the recent identification of a young patient in Denmark displaying homozygous $^{R25C}$PTH has opened avenues for observing the direct impacts of $^{R25C}$PTH within the human biological system (*Andersen et al., 2022*). The continual monitoring and observation of patients will contribute to a more profound comprehension of the long-term consequences associated with $^{R25C}$PTH exposure. This extensive observation is crucial in delineating the extended effects of this compound on individuals. Consequently, by conducting thorough investigations to confirm the potential bone anabolic effect of $^{R25C}$PTH, we hope to develop a novel bone anabolic agent with a targeted focus on the PTH1R.

## Materials and methods
### Plasmid construction
The coding sequences (CDS) of pre-pro-PTH and the mutated form, $^{R56C}$pre-pro-PTH[$^{R25C}$PTH], were amplified using primers containing the attB site (*Supplementary file 2*). These CDS fragments were obtained from pcDNA3.0-(hpre-pro-PTH)-IRES and pcDNA3.0-(h$^{R56C}$pre-pro-PTH)-IRES, which were used in the previous research conducted by *Lee et al., 2015*. Both CDS were introduced into donor vector pDONR223 with Gateway BP Clonase II Enzyme mix kit (Invitrogen, USA). Then, pre-pro-PTH and $^{R56C}$pre-pro-PTH were each cloned into pcDNA3.1-ccdB-3xFLAG-V5 with LR Gateway LR Clonase II Enzyme mix (Invitrogen, USA) to construct pcDNA3.1-(pre-pro-PTH)-3xFLAG-V5, and pcDNA3.1-($^{R56C}$pre-pro-PTH)-3xFLAG-V5. All experimental procedures were done with the manufacturer's

instruction. pDONR223 was a gift from Kim Lab (Roswell Park Comprehensive Cancer Center, USA), and pcDNA3.1-ccdB-3xFLAG-V5 was a gift from Taipale Lab (Donnelly Centre, University of Toronto, Canada).

## Cell lines used in this study

HEK293T cell line was a gift from the Lab of Viruses and Molecular Genome Engineering (Postech, South Korea). GP-2.3 cells which are derived from HEK293 cells (obtained from ATCC, ATCC# CRL-1573) was a gift from the Mass General Research Institute (Endocrine Unit, Massachusetts General Hospital and Harvard Medical School, USA). SGS-72 cells which are derived from human osteosarcoma cell line SaOS-2 (obtained from ATCC, ATCC# HTB-85) was a gift from the Mass General Research Institute (Endocrine Unit, Massachusetts General Hospital and Harvard Medical School, USA). We routinely perform mycoplasma detection using MycoStrip (Invivogen, Cat. No. rep-mys-10, Hong Kong) and have confirmed that mycoplasma is not detected in the cells we use in our experiments.

## Cell culture

All cell lines were grown at 37°C in 5% $CO_2$. HEK293T cells were cultured in Dulbecco's Modified Eagle Medium (Cytiva, HyClone DMEM/High glucose with L-Glutamine and HEPES, Cat No. SH30243.01, USA) with 10% FBS (Gibco, Fetal Bovine Serum, certified, Origin: USA, Cat No. 16000044, Lot No. 2522247RP, USA) and 1% penicillin-streptomycin (Cytiva, HyClone Penicillin-Streptomycin 100× solution, Cat No. SV30010, USA).

## Transfection

pcDNA3.1-(pre-pro-PTH)-3xFLAG-V5 and pcDNA3.1-($^{R56C}$pre-pro-PTH)-3xFLAG-V5 were each transfected into HEK293T with Lipofectamine 3000 Transfection Reagent (Invitrogen, Cat No. L3000001, USA) according to the manufacturer's instruction. After 48 hr of transfection, a culture medium was collected and used for western blot as a secreted protein sample. The rest of the cells were lysed by RIPA buffer (Thermo Scientific, RIPA Lysis and Extraction Buffer, Cat No. 89900, USA) following the manufacturer's instruction and used for western blot as total cell lysate sample.

## Western blot

Protein samples were prepared in two types, reduced sample and non-reduced sample. Reduced samples were a mixture of protein (secreted protein or cell lysate), sample buffer (Invitrogen, NuPAGE LDS Sample Buffer [4×], Cat No. NP0007, USA), with reducing agent (Invitrogen, 10× Bolt Sample Reducing Agent, Cat No. B0009, USA) and heated for 5 min at 95°C. Non-reduced samples were a mixture of protein, and sample buffer, without reducing agent, and not heated. The protein samples were loaded onto 4–12% Bis-Tris protein gels (GeneSTAR, StarPAGE Bis-Tris 4–12%, 15-well, Cat No. GPG4115), and ran with MOPS/SDS running buffer (GeneSTAR, 20× MOPS/SDS Running Buffer, Cat No. GMB0080). Transfer to the membrane was done by iBlot 2 Dry Blotting System (Invitrogen, iBlot 2 Gel Transfer Device, Cat No. IB21001, USA) with PVDF transfer stack (Invitrogen, iBlot 2 Transfer Stacks-PVDF-mini, Cat No. IB24002, USA) according to the manufacturer's instruction. The membranes were blocked for 1 hr at room temperature (RT) in 5% skim-milk solution in Tris-buffered saline (TBS; 20 mM Tris-base, 500 mM NaCl, pH 7.5) and then washed three times for 10 min each with Tris-buffered saline with 0.05% Tween-20 (TBST). The membranes were incubated with primary antibody for 1 hr at RT, then washed three times for 10 min each with TBST. If needed, the membranes were incubated with horseradish peroxidase (HRP)-conjugated secondary antibody for 1 hr at RT after primary antibody incubation, then washed three times for 10 min each with TBST. After washing, the membranes were rinsed and soaked in TBS. To develop a blot image, the membranes were treated with a chemiluminescent substrate solution (Merck Millipore, Immobilon ECL Ultra Western HRP Substrate, Cat No. WBULS0500, USA) according to the manufacturer's instruction. The blot images were obtained by LAS 4000 mini (Cytiva, ImageQuant LAS 4000 mini, USA). The dilution condition for the anti-FLAG with hHRP-conjugated antibody (Sigma-Aldrich, Monoclonal ANTI-FLAG M2-Peroxidase [HRP] antibody produced in mouse, Cat No. A8592, USA) was 1:2000. The dilution condition for the anti-HSP90 primary antibody (Santa Cruz Biotechnology Inc, HSP 90 Antibody [AC-16], Cat No. sc-101494, USA) was 1:5000. The dilution condition for the anti-mouse secondary antibody was 1:5000. Each antibody was diluted in TBST with 1% BSA solution.

## Proteasome inhibition assay

HEK293T cells were seeded in culture dishes at approximately 60% confluence, and they were allowed to grow for about 20–24 hr prior to transfection. The transfection of pcDNA3.1-(pre-pro-PTH)-3xFLAG-V5 and pcDNA3.1-($^{R56C}$pre-pro-PTH)-3xFLAG-V5 was conducted following the method mentioned earlier. After 24 hr of transfection, MG132, dissolved in DMSO, was added to the cells to achieve a final concentration of 10 μM. For the mock treatment, DMSO alone was added. The cells were then incubated for an additional 24 hr after MG132 treatment. Both the culture medium and cell lysate were prepared for western blot analysis to assess the restored protein levels. The western blot procedure was carried out as described in the previous section.

## Peptide synthesis and quantification

Human PTH(1–34), $^{R25C}$PTH(1–34), and dimeric $^{R25C}$PTH(1–34) were chemically synthesized by Anygen (Gwangju, Republic of Korea). The purity and mass of each peptide were analyzed by high-performance liquid chromatography and matrix-assisted laser desorption ionization-time of flight mass spectrometry in the manufacturer.

## PTH1R competition binding assay

The binding of PTH and its analogs to G protein-uncoupled PTH1R ($R^0$ conformation) and G protein-coupled PTH1R (RG conformation) was assessed using a competition method with membranes prepared from GP-2.3 cells (HEK-293 cells stably expressing the hPTH1R). For tracer radioligands, we utilized $^{125}$I-PTH(1–34) and $^{125}$I-MPTH(1–15). The unlabeled ligands tested were PTH(1–34), $^{R25C}$PTH(1–34), and dimeric $^{R25C}$PTH(1–34). Binding to the $R^0$ conformation was assessed using $^{125}$I-PTH(1–34) as the tracer radioligand, while binding to the RG conformation was assessed using $^{125}$I-MPTH(1–15). The addition of unlabeled ligands PTH(1–34), $^{R25C}$PTH(1–34), or dimeric $^{R25C}$PTH(1–34) caused dissociation of the tracer radioligand from each receptor if it had affinity to the receptors. Measurement of the dissociated ratio of the radioligand indicated the binding affinity between PTH1R and each unlabeled ligand. Each experiment was replicated four times.

## cAMP assay

To measure intracellular cAMP production, SGS-72 cells, derived from SaOS-2 cells and stably expressing the Glosensor cAMP reporter, were utilized to measure intracellular cAMP production. The detection of cAMP-dependent expression was performed using an Envision plate reader (PerkinElmer, Waltham, MA, USA), based on luciferase-based luminescence, as previously described by *Maeda et al., 2013*. Each measurement was replicated four times.

## Animal model used in the study

CD1 female mice were purchased from Charles River Laboratories (MA, USA), and all animal care and experimental procedures were conducted under the guidelines set by the Institutional Animal Care and Use Committee (IACUC) at Massachusetts General Hospital (MGH). The mice were housed in a specific pathogen-free environment, with 4–5 mice per cage, under a 12 hr light cycle at a temperature of 22 ± 2°C.

Eight-week-old C57BL/6N female mice were purchased from KOATECH (Gyeonggi-do, Republic of Korea) and stabilized for 2 weeks. All animal care and experimental procedures were conducted under the guidelines set by the Institutional Animal Care and Use Committees of Kyungpook National University (KNU-2021-0101). The mice were housed in a specific pathogen-free environment, with 4–5 mice per cage, under a 12 hr light cycle at 22 ± 2°C. They were provided with standard rodent chow and water ad libitum.

An OVX osteoporosis mouse model was established using 10-week-old C57BL/6N female mice. Following surgery, mice were divided into the following four groups (n=6 mice/group) as follows: Sham, OVX-control group, OVX+PTH (1–34) treated group (40 μg/kg/day), and OVX+dimeric $^{R25C}$PTH-treated group (40–80 μg/kg/day). OVX mice were allowed to recover for 4 weeks after surgery. Afterward, PTH (1–34) or $^{R25C}$PTH was injected subcutaneously five times a week for 4 weeks. μ-CT and histological analyses were performed on 4 groups at 18 weeks of age.

## Acute injections

The peptides PTH(1–34) and dimeric [R25C]PTH(1–34) were diluted in a solution comprising 0.05% Tween 80, 10 mM citric acid, and 150 mM NaCl at a pH of 5.0. Intravenous injections of these peptides were administered at doses ranging from 50 to 100 µg/kg into 9-week-old CD1 female mice. As a control, mice received only the vehicle. Levels of ionized calcium ($Ca^{2+}$) in the blood (n=6 mice/group) were measured at serial time points of pre-injection, 1, 2, 4, and 6 hr post-injection. Inorganic phosphate (Pi) in plasma (n=12 mice/group) and the excretion of Pi into urine (n=6 mice/group) were measured in samples obtained at serial time points of pre-injection, at 6 min, and 1, 2, and 6 hr post-injection.

## Calvarial injection mouse model

C57BL/6 male mice (8-week-old) were divided into the following three groups (n=6 mice/group): control, PTH (1–34) treated group (80 µg/kg/day) and [R25C]PTH-treated group (160 µg/kg/day). Subcutaneous injections of the respective drugs were administered once daily for 6 days. On the 16th day, the mice were sacrificed, and their bone tissues were harvested and fixed in 10% formaldehyde at 4°C. The fixed bone tissues were then decalcified in phosphate-buffered saline (PBS) (pH 7.4) containing 0.5 moles of ethylenediaminetetraacetic acid (EDTA). Following decalcification, the tissues were embedded in paraffin, and paraffinized tissues were sectioned to a thickness of 5–7 µm. Histological analysis was performed using the sectioned tissue slides stained with H&E. The area of new bone formation, which displays a more intense coloration compared to the existing bone tissue, was examined.

## µ-CT analysis

Mouse femurs were fixed in a 4% paraformaldehyde solution for 24 hr at 4°C. In µ-CT, we used the Quantum FX µ-CT (PerkinElmer, Waltham, MA, USA). The images were acquired at a 9.7 µm voxel resolution, with settings of 90 kV and 200 µA, a 10 mm field of view, and a 3 min exposure time. Serial cross-sectional images were reconstructed using the Analyze 12.0 software (Overland Park, KS, USA). To ensure consistent analysis, identical regions of interest (ROIs) were selected for the trabecular and cortical bones. Bone parameters and density were analyzed in the region between 0.3 and 1.755 mm (150 slices) from the bottom of the growth plate. Analysis of bone structure was performed using adaptive thresholding in CT Analyser. Thresholds for analysis were determined manually based on grayscale values for each experimental group: trabecular bone: 3000; cortical bone: 5000 for all samples. All bone parameters were evaluated according to the guidelines of the American Society for Bone and Mineral Research (*Bouxsein et al., 2010*).

## Three-point bending test

The left femur of the mouse was immersed in 0.9% NaCl solution, wrapped in gauze, and stored at −20°C until ready for a three-point bending test. In this test, we placed the mouse femurs horizontally with the anterior surface facing upward, centered on the supports, and the compressive force was applied vertically to the mid-shaft. The pressure sensor was positioned at a distance that allowed maximum allowable pressure (200 N) without interfering with the test (20.0 mm for the femur). A miniature material testing machine (Instron, MA, USA) was used for this test. The crosshead speed was decreased to 1 mm/min until failure. During the test, force-displacement data were collected to determine the maximum load and slope of the bones.

## Serum biochemistry analysis

Serum bone resorption and formation marker levels, specifically the C-terminal telopeptide of type I collagen (CTX) and procollagen type I N-terminal propeptide (P1NP), were assessed in mice from the Sham, OVX-control, PTH(1–34), and dimeric [R25C]PTH(1–34) groups by using the enzyme-linked immunosorbent assay (ELISA) according to the manufacturer's instructions. Additionally, their concentrations were determined using specific mouse CTX-1 and P1NP ELISA kits (Cloud Clone, Wuhan, China) respectively.

## Bone histological analyses

The tibiae were initially fixed in 4% paraformaldehyde at 4°C overnight. The following day, samples were decalcified using 10% EDTA solution (pH 7.4) for 4 weeks at 4°C. The decalcified tibiae were then

embedded in paraffin and sectioned at 3 µm thick. For TRAP staining, dehydrated paraffin sections were fixed in an acetone/ethanol mixture (1:1) for 1 min, followed by complete air-drying at RT for 20 min. Thereafter, the sections were immersed in TRAP reagent for 30 min at 37°C. In the histomorphometry analysis for TRAP staining, we used the primary spongiosa for the trabecular ROI because of the barely detectable bone in the secondary spongiosa of OVX model. Osteoclast surface per bone surface (Oc.S/BS) and osteoclast number per bone surface (Oc.N/BS) analysis followed the ASBMR guidelines (*Dempster et al., 2013*).

## Dynamic bone histomorphometric analysis

To conduct dynamic histomorphometry analysis, we injected the mice with 25 mg/kg body weight of calcein (Sigma-Aldrich) or Alizarin Red S (Sigma-Aldrich) intraperitoneally before sacrifice, with 3- or 10-day intervals between injections as previously described (*Lim et al., 2015*; *Jin et al., 2024*). Briefly, femurs or vertebrates were fixed in 4% paraformaldehyde solution for 24 hr at 4°C. The following day, the samples were washed with PBS solution and then dehydrated using a gradient of ethanol (50%, 70%, 85%, 90%, and 100%). Subsequently, we embedded the dehydrated femurs or vertebrates in methyl methacrylate (Sigma) to prepare resin blocks. The resin blocks were sectioned at 6 µm thick by using a Leica SP1600 microtome (Leica Microsystems, Germany). The fluorescence signals of calcein (green) and Alizarin Red S (red) from ROIs were captured using a fluorescence microscope (Leica, Wetzlar, Germany). For vertebral bone analysis, bone mineralization was evaluated by von Kossa staining. The sections were placed in 2-methoxyethyl acetate (Sigma-Aldrich) for 20 min, followed by rehydration with serial ethanol solutions (100%, 90%, 80%, 70%, and 50%) and distilled water for 2 min each. The sections were subsequently dipped in a 1% $AgNO_3$ solution (Sigma-Aldrich) for 5 min under ultraviolet light photons, washed in distilled water for 5 min, and dipped in 5% sodium thiosulfate solution for 5 min to remove nonspecific binding. Finally, we covered the sections with mounting solution and captured images by using a Leica microscope. The parameters of dynamic bone histomorphometry were analyzed using the Bioquant Osteo 2019ME program (Bioquant Osteo, Nashville, TN, USA).

## Statistical analysis

Statistical analysis was performed in GraphPad Prism 10.1.2. The data are presented as the mean ± standard error of the means (SEM). Statistically significant differences between the two groups were determined using an analysis of variance (ANOVA). A p-value less than 0.05 was considered statistically significant.

## Acknowledgements

This work was supported by the National Research Foundation of Korea (NRF) grant funded by the Korea government (MSIT) (2022R1A2C3006002 to SL; 2022R1F1A1074610 and 2022R1A4A1025913 to HL; RS-2023-00217798 and 2021R1A2C3003675 SYL; 2022R1A2C1006105 to J-YC), Gachon University Gil Medical Center (FRD2023-12 to SL), and Korea University Grants.

## Additional information

### Competing interests

Sihoon Lee: Reviewing editor, *eLife*. The other authors declare that no competing interests exist.

### Funding

| Funder | Grant reference number | Author |
| --- | --- | --- |
| National Research Foundation of Korea | 2022R1A2C3006002 | Sihoon Lee |
| Gachon University | FRD2023-12 | Sihoon Lee |
| National Research Foundation of Korea | 2022R1F1A1074610 | Hunsang Lee |

| Funder | Grant reference number | Author |
| --- | --- | --- |
| National Research Foundation of Korea | 2022R1A4A1025913 | Hunsang Lee |
| National Research Foundation of Korea | RS-2023-00217798 | Soo Young Lee |
| National Research Foundation of Korea | 2021R1A2C3003675 | Soo Young Lee |
| National Research Foundation of Korea | 2022R1A2C1006105 | Je-Yong Choi |
| Korea University | | Minsoo Noh Hunsang Lee |

The funders had no role in study design, data collection and interpretation, or the decision to submit the work for publication.

## Author contributions

Minsoo Noh, Xiangguo Che, Conceptualization, Resources, Formal analysis, Validation, Investigation, Visualization, Methodology, Writing – original draft, Writing – review and editing; Xian Jin, Conceptualization, Methodology; Dong-Kyo Lee, Doo Ri Park, Conceptualization, Resources, Investigation, Methodology; Hyun-Ju Kim, Conceptualization, Supervision, Methodology; Soo Young Lee, Conceptualization, Funding acquisition, Methodology; Hunsang Lee, Conceptualization, Supervision, Funding acquisition, Methodology, Writing – review and editing; Thomas J Gardella, Conceptualization, Resources, Investigation, Writing – review and editing; Je-Yong Choi, Conceptualization, Supervision, Investigation, Project administration, Writing – review and editing; Sihoon Lee, Conceptualization, Supervision, Funding acquisition, Methodology, Project administration, Writing – review and editing

## Author ORCIDs

Minsoo Noh ⓘ https://orcid.org/0000-0002-1239-3783
Sihoon Lee ⓘ https://orcid.org/0000-0002-9444-5849

## Ethics

CD1 female mice were purchased from Charles River Laboratories (Massachusetts, USA), and all animal care and experimental procedures were conducted under the guidelines set by the Institutional Animal Care and Use Committee (IACUC) at Massachusetts General Hospital (MGH). C57BL/6N female mice were purchased from KOATECH (Gyeonggi-do, Republic of Korea), and all animal care and experimental procedures were conducted under the guidelines set by the Institutional Animal Care and Use Committees of Kyungpook National University (KNU-2021-0101).

Reviewer #1 (Public review): https://doi.org/10.7554/eLife.97579.5.sa1
Author response https://doi.org/10.7554/eLife.97579.5.sa2

---

# Additional files

## Supplementary files

Supplementary file 1. Raw data set for statistical analysis. The numerical data used to generate *Figure 2A, B*; *Figure 3A–C*; *Figure 4B*; *Figure 5B–D, F*; *Figure 6B, D*.

Supplementary file 2. List of primers used in this study.

MDAR checklist

## Data availability

All data generated or analysed during this study are included in the manuscript and supporting files. *Supplementary file 1* contains the numerical data used to generate the figures.

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
