## [Editor Report · eLife Assessment]

This work investigates the functional difference between the most commonly expressed form of PTH, and a mutant form of PTH, identified in a patient with chronic hypocalcemia and hyperphosphatemia which characterizes hypoparathyroidism. The authors investigate the hypothesis that this mutant PTH assumes a dimeric form in vivo and serves anabolic functions in the bone. The data are **compelling** and the translational aspects are **fundamental** in understanding PTH-1 Receptor activation.

---

## [Referee Report · Reviewer #1 (Public review)]

Summary:

In this work, the authors investigate the functional difference between the most commonly expressed form of PTH, and a novel point mutation in PTH identified in a patient with chronic hypocalcemia and hyperphosphatemia. The value of this mutant form of PTH as a potential anabolic agent for bone is investigated alongside PTH(1-84), which is a current anabolic therapy. The authors have achieved the aims of the study.

Strengths:

The work is novel, as it describes the function of a novel, naturally occurring, variant of PTH in terms of its ability to dimerise, to lead to cAMP activation, to increase serum calcium, and its pharmacological action compared to normal PTH.

Comments on revisions: No further recommendations for revisions. Acceptable as the paper stands.

[Editors' note: the original reviews are here, https://doi.org/10.7554/eLife.97579.1.sa1]

---

## [Author Response]

The following is the authors’ response to the previous reviews.

**Public Reviews:**

**Reviewer #1 (Public review):**
Summary:In this work, the authors investigate the functional difference between the most commonly expressed form of PTH, and a novel point mutation in PTH identified in a patient with chronic hypocalcemia and hyperphosphatemia. The value of this mutant form of PTH as a potential anabolic agent for bone is investigated alongside PTH(1-84), which is a current anabolic therapy. The authors have achieved the aims of the study.Strengths:The work is novel, as it describes the function of a novel, naturally occurring, variant of PTH in terms of its ability to dimerise, to lead to cAMP activation, to increase serum calcium, and its pharmacological action compared to normal PTH.
**Recommendations for the authors:**
(1) In your response to the reviewers you included a figure. You said it was for the reviewers only. We are *not* including it here. Is that correct or should it be in the Public Reviews?

We apologize for any confusion and appreciate your thorough review. The phrase “data only for reviewers” was intended to indicate that the content was included in the revision based on reviewers’ comments, not in the main text (article). However, we acknowledge that this phrasing may be inappropriate. We are agree to make the figure included in the previous author response of the public reviews. Accordingly, we propose to revise the previous author response as follows:

- Remove "(data only for reviewers)".

- Correct the typo from "perosteal" to "periosteal".

- “Thank you for your comment. First, we ensured that the bones sampled during the experiment showed no defects, and we carefully separated the femur bones from the mice to preserve their integrity. In the 3-point bending test, PTH treatment significantly increased the maximum load of the femur bone compared to the OVX-control group. Additionally, the maximum load in the PTH treatment group was significantly greater than that observed in the PTH dimer group. Furthermore, structural factors influencing bone strength, such as the periosteal perimeter and the endocortical bone perimeter, were also increased in the PTH treatment group compared to the PTH dimer group.”

(2) Do you mean to always have R^0^ (have a superscript) and RG (never have a superscript) or should they be shown in the same way throughout your paper?

Thank you for your thorough review. Based on previous studies that addressed the conformation of PTH1R, R^0^ is typically shown with a superscript, while RG is not (Hoare et al., 2001; Dean et al., 2006; Okazaki et al., 2008). We have followed this notation and will ensure consistency throughout our paper.

Hoare, S. R., Gardella, T. J., & Usdin, T. B. (2001). Evaluating the signal transduction mechanism of the parathyroid hormone 1 receptor: effect of receptor-G-protein interaction on the ligand binding mechanism and receptor conformation. Journal of Biological Chemistry, 276(11), 7741-7753.

Dean, T., Linglart, A., Mahon, M. J., Bastepe, M., Jüppner, H., Potts Jr, J. T., & Gardella, T. J. (2006). Mechanisms of ligand binding to the parathyroid hormone (PTH)/PTH-related protein receptor: selectivity of a modified PTH (1–15) radioligand for GαS-coupled receptor conformations. Molecular endocrinology, 20(4), 931-943.

Okazaki, M., Ferrandon, S., Vilardaga, J. P., Bouxsein, M. L., Potts Jr, J. T., & Gardella, T. J. (2008). Prolonged signaling at the parathyroid hormone receptor by peptide ligands targeted to a specific receptor conformation. Proceedings of the National Academy of Sciences, 105(43), 16525-16530.

(3) The following grammatical and fact changes and word changes are requested.

We appreciate the thoughtful review and thank you for pointing out the grammatical, factual, and word changes required. We have carefully reviewed and addressed each of these corrections to ensure the paper's accuracy and readability.

We appreciate the reviewers' detailed and constructive reviews. We have addressed all the comments to improve the quality of our paper.